# Interculturalizing Religious Education—Mission Completed?

**Erna Zonne-Gätjens**

Fachhochschule für Interkulturelle Theologie Hermannsburg, D-29320 Südheide, Germany;
e.zonne@fh-hermannsburg.de

**Abstract:** In 1996 the German Länder started the 'mission' to interculturalize all subjects, including religious education (RE). Interculturalizing also applies for RE taught in conformity with the oldest model for RE. In so-called 'confessional RE' at state schools, it is the Catholic teacher who teaches children of several classes of the same year in one denominational RE group. The Protestant teacher teaches children whose parents ticked off "Protestant RE". How this model came into existence is displayed in a historical introduction of this chapter. However, a newer model called 'cooperative RE' is gaining popularity. In various schools there is ecumenical education by both Catholic and Protestant staff or multireligious education by Jewish, Christian, or Muslim teachers. New publications on this latter model have a focus on organizational matters, but also shed a light on interreligious learning. However, in this chapter the focus is on how intercultural issues are dealt with in the classroom within the first model. After all, confessional RE is still the standard and most common model in Germany. Therefore, this article will focus on Protestant confessional RE that is not organized in cooperation with Islamic, Jewish, or Catholic colleagues.

**Keywords:** confessional RE; history of RE; denominations; intercultural competence; intercultural theology; teacher biography; intrareligious plurality

## 1. Introduction

The Former Federal Minister for the Interior Wolfgang Schäuble stated in 2006 on the Islam Conference that "Islam is part of Germany and Europe". The current Federal President of Germany Dr. Frank-Walter Steinmaier highlighted, in September 2021 in his speech on the celebration of 60 years of "Guest Worker Agreements" with Turkey, Italy, Greece, Spain, etc., that Germany over the years has become a country with a migration background. The President praised the outcome. During this period, German culture has become more open.

An optimist might now suppose that the central pillar of cultural secondary socialization, the educational system, has also been interculturally opened. A precondition for interculturalizing of education is having staff at schools and universities that mirror the societal diversity as a team, that are trained in intercultural matters and hence are aware of the dynamics of (sub-)cultural differences. However, this is not enough. A framework like the curricula should set goals that do not go hand in hand with acquiring norms and values of the mainstream culture only. Indeed, the 1996 statement of the Standing Conference of Ministers of Education and Cultural Affairs of the Länder in the Federal Republic of Germany is fostering interculturalizing of all subjects (cf. Section 4).

Looking at the *status quo* of interculturalizing of education in Germany, RE is especially interesting. At the Faculty of Pedagogy, prospective RE teachers are currently introduced—alongside with future teachers of other subjects—to the pedagogy of diversity. However, in their internships they notice that RE in Germany is in most cases still a mono-confessional education. Some teacher trainees have noticed that during their matriculation procedure. Prospective RE teachers are not only enrolled in the Faculty of Pedagogy, but also take seminars that take place at separate faculties or institutes, e.g., for Catholic Theology or

Lutheran Theology. Hence, mono-theology academics mirror the standard model mono-confessional RE (Bauer 2019, p. 104). Moreover, most of the students of these separated Catholic and Lutheran theological faculties do not have a migration background but a German background.

Leaving the standard model of mono-confessional RE and introducing more opportunities for intercultural interreligious teaching and engaging staff of multi-religious background is not that simple, since history comes with a long and complex tradition of a very secured position of mainline churches and their respective RE courses at state schools in the German Constitution (cf. Section 2).

Many believe that in a mono-confessional RE the chances for interculturalizing are small since the model does not suggest plurality. However, in the last decades, pupils of different Christian (migrant) communities enter mono-confessional, e.g., Catholic, or Lutheran RE, because at their schools there are no Baptist, Methodist, etc., alternatives. But are RE teachers of mono-confessional RE equipped to recognize, to inform, and to intensify their beliefs and traditions? Is the mission of interculturalizing denominational RE accomplished? Analyzing curriculum documents, teacher interviews, and Lutheran Church publications will help to answer these questions by the end of this article.

## 2. Confessional RE in the Context of German State Schools

### 2.1. Christian Education in the Middle Ages and the Early Modern Era

During the Middle Ages the school system was in the hands of the church. The Reformation brought change. The territorial princes (*Landesherren*) were summoned to implement care for the Christian faith and church affairs (*cura religionis)* and to attend educational issues (Heinig 2018, p. 40). Founded in Martin Luther's understanding of justification by faith and in denial that personal faith is created by the Church, in Protestant territories everyone was supposed to have the competence to read the Bible and to understand the Gospel. Hence, all Protestant towns and cities were considered to maintain Bible-centered schools. Congregations should offer confirmation classes. Each male head of the Protestant household was assigned the daily duty to teach the catechism to his kin as well as the maids and his servants or workers (Kim et al. 2018, pp. 22–23). Later, Sunday school and Christian youth work entered the scene as well. Since the Peace of Augsburg (1555), following *cuius region*, *eius religion* (whose realm, his religion), the respective ruler chose the religion in his own territory. The peace settlement allowed nobles to decide what confession their kingdom would follow: Catholicism or Protestantism. "Germany was composed of denominationally homogenous regions or states. Within those regions, even the state-maintained schools were denominational. Consequently, Protestants would automatically attend Protestant schools, just as Catholics would attend Catholic schools" (Kim et al. 2018, p. 24).

### 2.2. Religious Education in the Modern Era

In 1803, Napoleon conquered territories on the left side of the Rhine. The German princes gained new territories on the right side of the Rhine. Former Church land and assets were confiscated but not compensated. However, the Churches were allowed to levy mandatory church taxes on believers. They were collected by the state and then forwarded to the authorities of the religious community—minus a commission.[1]

During the reorganization of Europe after the Napoleonic Wars, at the Congress of Vienna (1814–15), there was "( . . . ) relatively little attention to religious differences between the rulers and the restructured territories; for instance, Prussia ( . . . ) gained heavily Catholic territories, and Bavaria gained predominantly Protestant territories" (Haupt 2011, p. 146). Hence, "( . . . ) the developments during that time were not based on religious homogeneity within the territories. Increasing religious diversity—meaning at that time the presence of Catholics and Protestants—in the states led to an emphasis on equality, between the Christian denominations and among the individual citizens" (Haupt 2011, p. 146). Because of the new and restructured territories, in the respective states not only

new denominations but also new cultures were included. Accordingly, there was a certain influence on mainstream culture in the respective state.

From 1817 onwards, the State pledged to compensate the Churches for their losses suffered from the forced secularization of the clerical territories and church property during the Napoleonic Wars. These equalization payments involved, e.g., securing the salary of the pastors.[2]

After the Vatican Council proclaimed papal infallibility in 1870, chancellor Otto von Bismarck subjected the Catholic Church to state controls. The Catholic bureau in the Prussian Ministry of Culture was abolished. Catholic priests were forbidden to voice political opinions from the pulpit. All religious schools became subject to state inspection. Religious teachers were excluded from state schools. Strict state controls were placed over religious training and even over ecclesiastical appointments within the Church. Clergy on the Prussian territory were required to be German citizens who had studied theology at a German university and passed an academic state examination. The State obtained the veto over Church appointments. Dioceses that failed to comply with state regulations were cut off from state aid, and noncompliant clergy were exiled (Haupt 2011, p. 149).

### 2.3. Religious Education in the Weimar Republic

In the introduction it was indicated that, in comparison to other countries, there are German religious communities who have an extraordinary secure position that is fixed in the current Constitution. The paragraphs in the Constitution of 1949 were more or less "dusted of" after the Second World war and "copied and pasted" from the Weimar Constitution. Hence, the period of the Weimar Republic is very important for today's regulations.

In the beginning of the democratic Weimar Republic, regulations concerning old and established churches and religious communities that applied before 1919, were codified in the new Weimar Constitution (CWR). Moreover, some religious communities, like the Catholic and Lutheran Church and the Orthodox Jewish congregations, were entitled to the status as "corporation under public law" (Article 137, Section 5, subsection 1 CWR).

"Those religious communities which attain the status of a public corporation according to Article 137 (5) of the Weimar Constitution enjoy certain rights and privileges the other religious communities do not have" (Weiß and Adogome 2000, p. 52).

They are allowed (but not obliged) to levy taxes (Article 137 (6) CWR). As an entity of public law, they are not restricted to use private law but can operate public law. They are deprived of some public fees, e.g., the legal fees a claimant must pay during court proceedings (Weiß and Adogome 2000, p. 53).

The Lutheran and Catholic Churches in Germany enjoyed additional rights which were laid down in Concordats, i.e., special treaties between the federal states (*Länder*) and the Churches. "These treaties provide the establishment of Lutheran or Catholic theological departments at public universities, whereby the state refunds ninety percent of the overall costs, the obligation of the state to pay bishops and other church ministers, as well as some more financial assistances" (Weiß and Adogome 2000, pp. 55–56). Moreover, Article 138 (1) CWR enables the continuation of the equalization payments that started in 1817 (Weiß and Adogome 2000, pp. 55–56; see Note 2).

In the Weimar educational compromise (*Weimarer Schulkompromiss*) of 1919, the Churches and the Christian political parties—at least in theory—gave up the denominational school (*Bekenntnisschule)* as a mainstream school (*Regelschule*). In return, they were able to enforce a confessional RE at—from this time on—religiously mixed public schools (Bauer 2019, pp. 93–94). Teaching RE according to a mono-confessional model was installed and secured.

### 2.4. Religious Education during and after the Second World War

During the Nazi regime, Churches were subject to forcible-coordination (*Gleichschaltung*). The denominational school system was suppressed. Instead, the German common

school (*Deutsche Gemeinschaftsschule)* was introduced ([Mendl 2021](), pp. 518–19). However, according to the Concordat between the Holy See and the Nazi regime (*Reichskonkordat*) of 1933, Catholic schools could continue if the Catholic Church refrained from political intervention. The Protestant Nazis (*Deutsche Christen*) won the Church election in 1933 and the Evangelical Lutheran Church became the Church of State (*Reichskirche*). A minority, the so-called Confessing Church (*Bekennende Kirche*), was prosecuted. In 1936, the Protestant Nazis decided to remove the Old Testament from Thuringian school curricula. In Wuerttemberg schools, parts of the Old Testament were eliminated from RE in the year 1937.[3] Overall, RE had to be cooptated (*gleichgeschaltet*) with Nazi ideology.

Because of the suppression of the denominational school system and the construction of a German common school by the Nazi regime, after the Second World War there was an increase of confessional schools ([Mendl 2021](), pp. 518–19).

While drafting the new Constitution, Christian political parties strongly favored a codification of pro-church provisions that had been in force during the Weimar Republic ([Haupt 2011](), p. 152). Therefore, in the new constitution of 1949 called *Grundgesetz* (*GG*) there are a lot of cross-references to the CWR.

Some provisions from the CWR were broadened. Hence, not only public corporations, like the Catholic Church, Lutheran Church, and Orthodox Jewish community are entitled to teach religious education in public schools. According to a popular interpretation of the new Constitution of 1949, Article 7 (3) GG, any religious community is allowed to teach RE if they meet and fulfil the same conditions as those communities which are public corporations. Their constitution and membership strength must offer an assurance of their permanency. Religious communities are allowed to participate in RE at state schools if they are public corporations or could become one if they applied for legal transformation into a public corporation ([Weiß and Adogome 2000](), pp. 54–55). However, religious communities that do not meet these prerequisites are not entitled to teach their beliefs in public schools. Hence, there were court proceedings of several religious communities that issued a complaint after having been denied the entitlement of carrying out RE at public schools. In the following, the Federal Administrative Court formulated criteria that religious communities must fulfil before they can account for RE in public schools[4]: they do not have to be public corporations but must fulfil analogue requirements. A religious community must accomplish identity-establishing tasks for its members. It must be in the position to determine the principles of its religion or denomination on behalf of their members towards the state. It must warrant a durable existence and organizational stability by statues and bodies. It must refrain from foreign governance ([Bauer 2019](), pp. 98–99).[5]

In congruence with the Weimar Constitution, the Constitution of 1949 displays that religion in Germany is a public matter. Distancing itself from laicistic France, the German State is indeed neutral in religious matters but sees itself in need to cooperate with religious communities in cases where religious or theological competences are required. By this means, the State fosters positive freedom of religion of the believers of one confession and at the same time the negative freedom of religion of all others ([Heinig 2018](), pp. 40–41). Because of its neutrality, the State can neither determine what a religion consists of nor how it is imparted. It can neither prefer a certain religion nor pre-set religious principles but can facilitate religious freedom as freedom to a certain confession. For the curriculum of confessional RE, the neutral state relies on concrete organized religious communities to display confessional contents ([Bauer 2019](), p. 98).

### 2.5. Religious Education in the 1960s until Present

Until the beginning of the 1960s, many public schools in Western Germany were denominational schools (*Konfessionsschulen*) and offered RE in line with their confession. At the end of the 1960s, the preferences of parents and politicians changed, and mixed non-denominational schools (*Gemeinschaftsschulen*) were established ([Heinig 2018](), pp. 40–41). Reasons for this shift included the increasing heterogeneity that was caused by the wave of post-war refugees and by mobility connected to industrialization and urbanization ([Kim

et al. 2018, p. 24). Since then, Protestant and Catholic (and sometimes Jewish) RE classes are the only "remnants of denominational schooling" (Kim et al. 2018, p. 24).

With almost a state monopoly on education, confessional RE as regular school subject warrants individual religious freedom. The state is obliged to train and hire qualified teachers (Bauer 2019, p. 94). Confessional RE has a very robust place in the German educational system as it is the sole subject that is regulated in the Constitution. The RE teachers normally have a calling of their religious community,[6] e.g., the Catholic RE teachers have a *missio canonica*, the Protestant RE teachers a *vocatio*.[7] This exceptional position protects RE from laicistic wishes for elimination and guards this subject from organizational or financial predicaments. On the other hand, it is also the reason for special requirements concerning its formal framework and its configuration of contents and didactics (Bauer 2019, p. 93). For parents who want their children nurtured in a certain denominational tradition, the solid position of confessional RE is conductive to acceptance of public schooling and prevents raising numbers of confessional private schools. Hence, the number of church private schools is, in comparison to the laicistic France, relatively small (Heinig 2018, p. 46).

This is still valid in the 21st century. "More or less every Protestant child attends Protestant religious education at school and continues to do so into middle or late adolescence, i.e., on a weekly basis with usually two hours per week for a total of more than ten school years" (Kim et al. 2018, p. 54). Even though more and more German pupils are raised in a secular way at home, the long-running encounter of a confessional culture in a school setting has survived secularization.

According to Article 7 Section 3 of the Constitution, RE is a regular school subject. Without prejudice to state supervision, RE is taught in accordance with the principles of the respective religious communities. According to Article 7 Sections 2 and 3, (non-)participation is decided by the legal guardians of the child, whereas teachers can decline to administer RE (Bauer 2019, p. 93).[8] Hence, confessional RE is an obligation for the public school, but this is not the case for the individual pupil or teacher (Bauer 2019, p. 94).

Currently, in most federal states of Germany, confessional RE is the standard model in which, ideally, the denomination of teacher, pupils, and RE contents are identical (Bauer 2019, p. 104). As personal identity formation and the emergence of truth in faith is seen as a mediation process, the religious recognizability of the RE teacher is essential (Wißmann 2019, p. 81).

As secularization also applies for the minority of baptized pupils, most are not very familiar with "their" Church. Hence, it is mainly the teacher who brings in the Protestant perspective in RE classes (Schröder 2021, pp. 82–83).

RE teachers are not only appointed by the State but are also officially assigned to this service by the respective religious communities with the Protestant *vocatio*, the Catholic *missio canonica*, the Jewish, *ischur*, the Muslim *idschaza*, and the Alevitic *risalik*. As official representatives, they must have a rooted and living faith in connection to and in congruence with the expectations of the religious community that has commissioned them.

The Evangelical Lutheran Church mandates the teacher by conferring the official church approval for teaching. Accepting the *vocatio*, Protestant RE teachers express their consensus with the principles of the Protestant Church.[9] These principles are not seen as doctrines but as means for orientation (Heger 2021, p. 145). Protestant RE teachers should make Christian faith manifest and plausible for the interpretation of existential issues (Heger 2021, p. 147).

The Protestant Church supports, encourages, and fosters her RE teachers by providing teaching materials, media for lessons, and ongoing training by the Church's Institutes for Religious Education (*Religionspädagogische Institute*). The Church also gives orientation in its declarations concerning current issues, such as inclusion, pupils with a refugee background, etc. However, the Church acts rather low-key when it comes to her supervision and control rights (Schröder 2021, pp. 82–83).

According to ecclesiastical law, the task of Catholic RE teachers is part of ministry. Local ordinaries in the context of shepherding must appoint such teachers who distinguish themselves by Catholic orthodoxy (*Rechtgläubigkeit*) and by Christian life.[10] Hence, Catholic RE teachers should not only be formal members of the Catholic Church but practice Catholic morality in their personal lives and actively participate in the local parish. They have a duty to transmit Catholic belief in accordance with doctrine (Heger 2021, p. 145).

Nonetheless, RE teachers should not be misunderstood as the extended arm or as the missionaries of the respective religious community in state schools. They rather act in a freedom that implies connection to their religious community (*in verbundener Freiheit zur Kirche*). RE is rendered discursive. RE teachers are expected to take a critical but loyal position towards the respective religious tradition. Through this, they become authentic witnesses of the faith and representatives of their religious community (cf. Heger 2021, p. 144).

A core element of the constitutional concept of RE is its confessionalism (*Bekenntnishaftigkeit*). In 1987, the Federal Administrative Court stated in its judgement that RE must ensure positive confessionalism and denominational dependence.[11] Its task is to inseminate religious truth of the responsible religious community as existing verity.[12] Hence, RE is in line with the Constitution if it is a positional RE and not a neutral subject like religious studies (Bauer 2019, p. 109). This implies a concentration on and reflective starting point from one confessional culture.

The legitimacy of RE is the providing of a spiritual home and a specific religious nurturing that enables identity formation (Wißmann 2019, p. 80). The identity formation process ideally binds the pupils to the responsible religious community by the confessional contents of its RE (Bauer 2019, p. 110). It is the fundamental right of a member of a certain denomination to receive transmission of religious truths cq. the right to receive spiritual nurturing in confessional RE (Wißmann 2019, p. 81).

In the last two decades, in certain Federal States of Germany, the number of religious communities offering their confessional RE has risen. In North Rhine Westphalia, there are currently eight and in Hessia thirteen different subjects of confessional divided RE (Bauer 2019, pp. 167–68).[13]

The degree of organization might be a reason for the fact that—although there are more than four million Muslims living in Germany—there still is no nation-wide Islamic RE (Kim et al. 2018, p. 62).[14] Due to the long-term cooperation between state and churches, there are established structures to determine guidelines for curricula of Protestant and Catholic RE. Those churches are large enough to appoint their own personnel, e.g., for the correspondence of with the Minister of Cultural and Educational Affairs. Other religious communities with lesser resources do not have this possibility. Decentralization of Islamic associations causes a smaller inner structure of administration, which is a disadvantage (Meyer 2019, p. 53).

The organizational level of several Islamic associations does not meet the prerequisites for a potential legal transformation into a public corporation. Also, the authority of some of the umbrella organizations over mosques is questionable. In most cases, it is not clear for whom the umbrella organizations speak and what the binding nature of their statements are. The Higher Administrative Court in Münster decided that the Central Court of the Muslims in Germany and the Islamic Council do not constitute religious communities in legal terms. Although the federal state Hessia cooperates with the tightly organized DITIB, other federal states decided against this cooperation as DITIB is under supervision of the Turkish Religious Authority *Diyanet* (Heinig 2018, p. 42).

To circumvent, North Rhine Westphalia and Lower-Saxony have integrated several umbrella organizations in an advisory board (*Beirat*). The advisory board exercises the participation rights as if it were a classic religious community. Not only representatives of umbrella organizations are active in the advisory boards, but also additional experts appointed by the State in agreement with the umbrella organizations (Heinig 2018, p. 43).

However, the neutrality of the state might be at stake as it has quite some power while choosing umbrella organizations and appointing additional experts (Heinig 2018, p. 43).

Denominational RE taught in a mono-confessional group of pupils and administered by one specific religious community has a denominational RE curriculum that aims at the nurturing of confessional truth. Mainly focusing on practices of one confessional culture, e.g., the Lutheran Mainline Church, other religions are looked upon from a confessional perspective. It is a learning about other faiths (Bauer 2019, p. 104).

In some schools, for different groups of Protestant, Catholic, Islamic, and other confessional RE subjects, every year there is/are project day(s) of religious encounter (Bauer 2019, p. 105). During these project days, there is a transparency of a certain identity relationship between individuals and a religion. It is mentioned what is mine and yours, my own or different—possibly with gradual differences (Bauer 2019, p. 112).

Due to secularization, the religious identification of pupils is often seemingly weak, and in confessional RE it is seen as something that must be fostered and not overshadowed by an unidentified other. The historic-cultural bondage with a certain confession is treasured in denominational RE (Bauer 2019, p. 157). However, mono-confessional RE can allow pupils of other confessions or of no confession to take part as 'guests' (cf. Pemsel-Maier 2019, p. 87).[15]

### 2.6. Special Arrangements in Berlin and Bremen

There is a special provision in the Constitution (Article 141 GG) concerning the federal states Bremen and Berlin. In Bremen, "Biblical History on general Christian Basis" is offered. In Berlin, religious communities are the providers, administrators, and supervisors of RE (Bauer 2019, p. 93). It is not a regular subject but an elective with separate enrolment for the school year with two hours per week in the Berlin curriculum. There is no mark on the report card of the pupil in Berlin. Hence, the subject is not relevant for moving up to the next grade. The RE educators are paid by the religious communities. The religious communities are refinanced by state subventions (Häusler 2020, p. 80). A permit according to §13 of the education act of the federal state of Berlin to administer RE at public schools in Berlin is given to the Alevite Community Berlin, the Buddhist Society Berlin, the Archdiocese Berlin, the Protestant Church Berlin-Brandenburg-schlesische Oberlausitz (EKBO), the Humanist Union Berlin-Brandenburg, the Islamic Federation Berlin, the Jewish community Berlin, and the Syrian orthodox church Mor Afrem (Häusler 2020, p. 80). In Berlin, from all aforementioned elective subjects, Protestant RE has the highest participation rate. In the school year 2018/19 it reached 21.6% of all pupils of Berlin schools of general education. Catholic RE reached 6.7% of all pupils of Berlin schools of general education (Häusler 2020, p. 81). Humanistic Life Skills reached 18.3% of all pupils of Berlin schools of general education (Häusler 2020, p. 86). Islamic RE reached 1.5% of all pupils of Berlin schools of general education (Häusler 2020, p. 84). Jewish RE reached 0.3% of all pupils of Berlin schools of general education (Häusler 2020, p. 85). Alevite RE reached 0.1% of all pupils of Berlin schools of general education (Häusler 2020, p. 85). In addition, since school year 2006/07, Ethics is obligatory and a regular school subject with two hours for all pupils of Berlin schools of general education in grades 7–10 (Häusler 2020, p. 87).

## 3. Cooperative RE with Ecumenical and Multireligious Teams
### 3.1. Juridical Perspective on Turning Denominational RE into Cooperative RE

A plausible hypothesis would be that in Cooperative RE, e.g., teaching with an ecumenical team, there will be more room for interculturalizing. However, from a juridical perspective, this is not an easy road to take. Many scholars have searched for possibilities and found legal openings, but also juridical barriers on their way.

Even though separation along confessional lines had a long tradition, in 1987, the Federal Constitutional Court emphasized that the term RE is not fixed in every aspect.[16] As the rest of the constitutional contents, the concept must be opened to suit the times ensuring the solutions of time-oriented and changeable problems (see Note 12). If the

understanding of RE changes on the side of the religious communities, the religiously neutral state must tolerate changed forms of RE. However, this does not apply for every random comprehension. According to the Federal Constitutional Court, the freedom of legal arrangement is confined by the constitutional term RE. Hence, it depends if the new form of RE still meets the constitutional criteria of Article 7 Section 3 GG. The new RE must still be administered in confessional positivity and denominational dependency (Bauer 2019, pp. 96–97) (see Note 12).

Instead of the CWR formulation of 1919 that was put in singular, the Constitution of 1949, Article 7 Abs. 3 GG, speaks in plural and demands a RE in agreement with the principles of the respective religious communities. Hence, if a school assembles a mixed group that is taught in consort with the doctrines of their corresponding congregations, there is no constitutional obligation to return to separate denominational groups of pupils (Bauer 2019, p. 96).

However, some federal states in Germany have additional regulations in their state constitutions, educational acts, and contractual agreements with religious communities. North Rhine Westphalia and Bavaria codified the separation upon denomination (Bauer 2019, p. 96). Also, Rhineland and Hessia stick to mono-confessional RE (Rothgangel 2020, p. 447).

Some federal states of Germany made other statutory provisions. In the Federal State of Hamburg, RE must be administered in the spirit of dignity and tolerance towards other confessions and worldviews (Bauer 2019, p. 96).[17] Accordingly, the Hamburger attempt to turn confessional RE into RE taught in the regular class—without rearranging pupils from different classes into separate confessional RE groups—caused a lot of discussion nationwide and was cause for the Lutheran Church of Northern Germany (*Nordkirche*) to commission an expert report of Wißmann (2019).

It was clarified that in a mixed group of pupils of different denominations, religions, and worldviews, certain parameters for teaching RE must be met (Wißmann 2019). The religious content of RE must be determined by the religious communities. The state must remain religiously neutral. Guidelines for the contents of RE must secure its confessionalism. If RE leaves out religious truth the state will take the obligation to structure a public schooling subject like Ethics or World Religions (cf. Wißmann 2019, p. 54).

In a mixed group setting, it must be warranted that representatives agree upon what is called a religious truth. However, in most Federal States of Germany, it is not clarified which respective religious umbrella organization can determine the Islamic truths, Buddhist doctrine, and Hindu principles, etc., for their respective religious communities and members (Bauer 2019, p. 99). There is still a lack of institutionalized collaboration of the office holders of the involved religious communities that formally share responsibility for cooperative RE with ecumenical and multireligious teams (Bauer 2019, p. 100).

Hence, it is fair to say that cooperative RE is often ecumenical RE. However, it is not clear if this specific model of RE is chosen—only or mainly—for its openness for interculturalizing RE.

### 3.2. Ecumenical RE

There is a more pragmatic argument for cooperative RE with ecumenical teams: secularization. Due to a rising number of pupils who belong to neither the Catholic nor the Protestant Church, more and more schools establish confessional cooperative RE, e.g., Wuerttemberg since 1993, Lower Saxony since 1998, Berlin and North Rhine Westphalia since 2018/2019 (Rothgangel 2020, p. 446; Simojoki 2020, p. 189).

Increasingly, due to secularization and multi-religiosity, schools cannot compile from one cohort separate traditional Catholic and Protestant RE classes, as one single denominational group should have—in most Federal States—an average of at least eight pupils (Simojoki 2020, p. 190; Heinig 2018, p. 43).

According to the Standing Conference of the Ministers of Education and Cultural Affairs of the Federal States, although two out of three pupils from grade 1–10 had sep-

arate Protestant or Catholic RE in the school year 2015/16, already 5% of all pupils had cooperative RE in either an ecumenical or multireligious setting (Heinig 2018, pp. 41–42).

In the case of cooperative RE, the alternative of mono-confessional RE is not always offered. However, it is a constitutional issue if pupils are compelled to participate in the RE of another denomination and their right to have RE in their own confession is undermined (Bauer 2019, p. 106).

Cooperative RE in Lower Saxony and Wuerttemberg often means that in principle there is still separate confessional RE. However, pupils by turns take part in RE of the other denomination. The latter does not lose its specific confessional character (Bauer 2019, p. 106). On the report card of a Protestant pupil, it can be stated that he or she visited, in a certain period, Catholic RE. In Wuerttemberg it is the confession of the teacher that determines the confessionalism of the RE (Bauer 2019, p. 106). The teacher warrants that the RE is taught in line with the religious principles of his/her own religious community. Ideally, by taking a transparent denominational position his/herself, the teacher fosters the identity formation of the pupils (Bauer 2019, p. 115). It is unclear, however, if all RE teachers have the ability to do that. In some federal states, RE teachers often teach outside their subject area. In Schleswig-Holstein, approximately 50% of teachers of Protestant RE teach without qualification for this subject (Pohl-Patalong 2020, p. 405).

Curricula are enacted by the individual religious community. However, due to coordination of planning, different religious communities can ensure a certain parallelism in and compatibility of their respective RE. Hence, after one term of Catholic RE, the Protestant pupil might visit Protestant RE again without large knowledge gaps (Bauer 2019, p. 106).

A clarification of the respective relationship towards the other religions by the representatives of the religious communities is required before starting cooperative RE (Bauer 2019, p. 101). Perspectivity is considered a key for cooperative RE. A religious community warrants that the curricula cover its own religious contents for pupils who are its members. At the same time, it must be safeguarded that pupils who belong to other denominations can recognize the content as distinct from their own faith (Bauer 2019, p. 102).

For pupils, it is rather easy to identify cultural aspects of a certain denomination, e.g., liturgy, songs, prayers, rituals, icons. However, distinguishing differences in doctrine, e.g., sin and grace, nature of human being, ecclesiology, is more difficult for them and must be made transparent by the teacher in categorizing "own and other" (Pemsel-Maier 2019, pp. 87–88; Bauer 2019, p. 114).

Interculturalizing of ecumenical RE is also limited by schoolbooks that include mainly two mainstream confessional cultures: Catholic and Lutheran.

### 3.3. RE for All 1.0 and RE for All 2.0

In 1995, the discussion group Interreligious RE started to meet in Hamburg. From this circle a project called 'RE for All in Protestant Responsibility' emerged. The situation in Hamburg is special. After the second world war, the Catholic Church decided to refrain from teaching RE at public schools in the Free and Hanseatic City of Hamburg. Instead, within the new established schools in ownership of the Catholic Church, confessional Catholic RE was administrated (Härle 2019, p. V).

RE for All at state schools in Hamburg is characterized by timely parallelism of denominational instruction and education about other confessions (Bauer 2019, p. 102).

Pupils are not divided into separate denominational RE groups but participate in one common RE that in theory is officially administered by several religious communities as providers. In this model, a plausible hypothesis is that there will be more room for interculturalizing processes as well as interreligious encounter and dialogue. However, currently it is the reality that only the Lutheran Church Hamburg is in charge. Hence, Protestant teachers have a determining influence. This affects other religious communities (Rothgangel 2020, p. 447). However, the multireligious curricula and learning materials were developed under informal participation of members of other religious communities (Bauer 2019, p. 107).

In the near future, it is intended that not only the Lutheran Church Hamburg installs teachers with the *vocation,* but other religious communities appoint teachers for RE for All as well. Jewish teachers officially appointed by the Hamburger Jewish Community should receive an *ischur*, Muslim teachers an *idschaza*, Alevite teachers a *risalik* not only for mono-confessional RE but also for RE for All. This official appointment would seal and legitimize RE for All. Called and authorized by their respective communities, the RE teachers should be authentic witnesses of their own faith (Härle 2019, pp. 136–37).

Hence, in future, a teacher of the revised RE for All 2.0 should be able to introduce pupils into the principles of religious communities. As a representative of his/her own religious community, the RE teacher should instruct in doctrine according to his/her calling, but also speak about principles of other religious communities, warrant a respectful handling of unfamiliar convictions, and moderate classroom dialogue about and between different beliefs (Härle 2019, p. 139). Distinguishing these different roles of the RE teachers will challenge both pupils and staff (Härle 2019, p. 140). Moreover, an authorized linking of several denominational perspectives of pupils by one RE teacher with an exclusive calling of a singular religious community is not possible as one teacher cannot have both *vocatio,* and *ischur*, and *idschaza*, and *risalik*, etc., and team teaching would be very expensive (cf. Bauer 2019, p. 108).

The presupposition is that teaching all pupils in one group means different religions and different cultures which enhances interculturalizing of RE. However, teachers with a calling of a particular religious community might find themselves in the dilemma of opposing assertions that cannot be positively linked to religious truth of his/her own religious community. However, according to the Constitution, RE teachers should transmit the verity of the religious community that has appointed them—indeed as the certain and valid Truth. Otherwise, RE would turn into the unconstitutional subject religious studies with a neutral display and comparison of contradicting doctrines (Härle 2019, pp. XI–XII).

An RE teacher with a *vocatio* of the Protestant Church can neither teach into Catholic, nor Jewish, Islamic, nor Alevite Truths, and vice versa (Härle 2019, pp. XIII and 1).

However, the RE teacher that is appointed by a particular community will teach contents related to other confessions. Hence, all religious communities must approve the employment of this individual teacher because of his/her personal abilities to realize the basic concept of RE for All including perspectivity and dialogicity and appropriate consideration of other confessions while preventing fundamentalistic overpowering (Bauer 2019, p. 103).

Since 2012, representatives of the board of education and religious communities, especially the Islamic umbrella organizations DITIB Hamburg and the council of Islamic communities SCHURA as well as the Association of Alevite community in Germany, have developed concepts and guidelines (Bauer 2019, p. 108; Härle 2019, pp. 35, 62, 65). Other religious communities remain skeptical.

According to the Archdiocese Hamburg, Catholic RE teachers administer confessional Catholic RE.[18] To prevent impairment of denominational homogeneity, Hamburger pupils who do not belong to the Catholic Church are not allowed to participate in Catholic RE (Härle 2019, p. 30). The Archdiocese Hamburg is currently examining, in a pilot project with the board of education and the Protestant Church of Northern Germany, if it is possible in a renewed RE for All 2.0 to authentically represent Catholicism by Catholic RE teachers with *missio canonica* (Kemnitzer and Roser 2021, p. 6; Bauer 2021, p. 40).

Concerning RE for All 2.0, it is not only the Catholic Church but also the Jewish Community in Hamburg that is standoffish (Härle 2019, p. 34). Most Hamburger orthodox Jewish pupils visit the local Jewish Private School. Pupils who live quite a distance from this school or are more liberal, can visit public school on daily basis, and have the possibility to visit the Jewish Private School only for special RE for external Jewish pupils (Schwarz 2021, p. 196). However, many persons associated with the Jewish community in Hamburg are secular and do not comply with the orthodox Jewish laws like the *kashrut*. The majority comes from the former Soviet Union where they had to assimilate and, es-

pecially under the Stalin regime, could not practice their faith. These members have only vague memories of how Jewish festivities are celebrated. They have only rudimentary knowledge of how Hebrew prayers and blessings are spoken or how rules in accordance with Jewish laws should be followed (Anusiewicz-Baer 2021, p. 125). Hence, it is highly questionable how teachers in RE for All will associate and appropriate these children to Judaism, as they are—according to Jewish law—not formal members of the Jewish community. When it comes to identity processes, there might be ascriptions that become a primer for identity formation without having practiced Jewish belief before or without having the possibility to simultaneously acquire the practice of Jewish faith. Hence, without additional visiting either of the external classes at the Jewish Private School or of RE in the synagogue, RE for All 2.0 is looked upon rather skeptically by the Jewish community (Anusiewicz-Baer 2021, p. 136). Instead, securing a monocultural orthodox nurturing is preferred by the Jewish community in Hamburg. Compared with what is at stake—Jewish identity—interculturalizing RE is seen as less important.

### 3.4. Confessional RE Revisited

In conclusion, mono-confessional RE remains the main model of RE. Possibilities of interculturalizing this RE seem limited in comparison with the also meagre results of the other models. However, there are possibilities, also for denominational RE. It is important to highlight the new pedagogical curricular for teacher training including Intercultural Education. Also, it still has to be investigated, in how far Intercultural Theology is a discipline that teachers have encountered during theological seminars at university and if it helps them with interculturalizing RE in state schools.

## 4. Intercultural Learning

### 4.1. Interculturalizing Education in Germany

In 1996, the Standing Conference of the Ministers of Education and Cultural Affairs of the *Länder* in the Federal Republic of Germany stated that schools must foster the acquisition of intercultural competences in all subjects and extracurricular activities. The reference to all subject means including RE (Kultusministerkonferenz 1996).

The adjective 'intercultural' in the statement of the Ministers of Education and Cultural Affairs neither stresses the parallel existence of cultures nor emphasizes the often negatively connoted immigration as much as the attribute 'multicultural' does. An intercultural school actively uses the presence of multicultural variety to positively shape its pedagogical actions (Gogolin and Krüger-Potratz 2020, p. 153).

According to the ninth of "Ten theses on religious education" of the Evangelical Church in Germany, published in 2006, RE plays an important role in the development of school programs. In her explanation of this thesis, the Church states that global learning is of "fundamental concern" and that "fortunately" there is an "increasing" number of school profiles and programs with clear references to intercultural learning and interreligious communication.

These programs follow the leading perspective of intercultural pedagogy, the idea of a multicultural society. This notion is based on the principle of recognition, particularly also of linguistic or religious diversity, while safeguarding both equality of chances regardless of background and inclusion into societal subsystems (Auernheimer 2016, pp. 19–20). Intercultural education aims at altering of attitudes, creating sensibility for asymmetries in power, and awareness of possible collective experiences of discrimination, but also the acquisition of knowledge and abilities, e.g., capacity to change perspectives, to enter into intercultural dialogue, and to effectively resolve normative differences (Auernheimer 2016, p. 20; Auernheimer 2018).

Interculturality emphasizes the purposeful interaction between cultures or rather between pupils who feel they belong to certain cultures or are allocated to those cultures. Recently, due to multiple contextual influences on an individual pupil causing a blend of social elements, there is more emphasis on hybrid identities as purity and exclusivity of

distinct cultural identities have become questionable ([Mecheril 2019](), p. 311). Intercultural pedagogy had to be reformed since the emphasis on intercultural competences may lead to an unintentional but undesirable culturalization: a misattribution following a false supposition that this pupil has, because of his nationality, ethnicity, or religion, a certain habitus, and a specific form of verbal and non-verbal communication—leaving out all possibilities for diversity within. Therefore, in the 21st century, instead of intercultural pedagogy, the concept of a pedagogy of diversity (*Pädagogik der Vielfalt*) is often used as an alternative to resolutely broaden the scope toward all appearances of diversity ([Allemann-Ghionda 2021]()). Here, the recognition of plurality of differences for individual and social identity structures is fundamental ([Auernheimer 2016](), p. 130). This diversity includes social status, sex, gender, sexual orientation, ethnicity, nationality, age, language, religion, mental health, disability, regionality, etc. The intention of the diversity concept is the recognition of the individual right to choose an autonomous way of life in context of current pluralization of designs of life and societal power relationships, and inequalities ([Auernheimer 2016](), p. 130). The concept of a pedagogy of diversity aims to acquire self-esteem, to recognize and to incite curiosity in the Other, to experience mutual learning, to pay attention to transitions, to plurality of differences, and to their intersections, but also to realize crossing relationships of dominance (e.g., girl with German ID—boy with foreign nationality) and to be sensible for societal and economic conditions ([Auernheimer 2016](), p. 131).

Following the pedagogy of diversity, a teacher monitors the handling of disconcertment of other—often very individualized—norms and cultural practices and guides pupils to an appreciation of individually meaningful symbols and rituals living up to the postulate of recognition. A teacher must be sensitive to differences but should not suggest differences. Also, it should be clarified that there are individual limits for understanding as well ([Auernheimer 2016](), p. 135).

### 4.2. Interculturalizing RE from the Perspective of Intercultural Theology

Prospective RE teachers in German do not only enroll in the pedagogical faculty. Next to pedagogics and "their" theological subject, they must study also one—sometimes two—other subjects like Mathematics, Biology, etc. Prior to starting their academic study, they must also decide if they matriculate in either Catholic, Protestant, Jewish, or Islamic theology. The separation into different theologies at university is in line with the separate confessional RE courses in schools.

However, all prospective teachers, including future RE teachers, are prepared for intercultural education in seminars at the pedagogical faculty (cf. Section [4.1]()). Hence, a certain influence coming from general pedagogy can be presumed and might inspire teacher training students to experiment with interculturalizing RE.

Further motivation for interculturalizing RE can be found in intercultural pedagogy and pedagogy of diversity and in the appeal of the Ministers of Education and Cultural Affairs in the year 1996 to interculturalize all school subjects. In Germany, the pedagogy of diversity might be the main line of thought for religious educators that graduated in the end of the 20th or in the first decades of the 21st century, because during their BA and MA studies they have followed obligatory studies—including lectures connected to the topic "heterogeneity"—in the department of education.

During their studies, of course, as a student they have also visited lectures in the respective Theological Faculty of their University. Hence, as RE as an academic discipline belongs to Theology, there is also another impetus for interculturalizing: Intercultural Theology. At some, but certainly not at all, faculties for Lutheran Theology and Catholic Theology there is either a program or a Chair for Intercultural Theology.

Even though the term "intercultural" was derived from linguistics and pedagogy and put to theological use in the 1970s by missiologists working in European (and secular) universities, Intercultural Theology should not be misunderstood as "a tactical update of the toolbox of missionary theology" ([Ustorf 2008](), p. 230). Intercultural Theology grew out

of missiology, religious studies, and ecumenical theology (Ustorf 2008, p. 233). Focusing on the revitalization of non-Western Christianity against the tide of secularism and riding on the wave of globalization of Christianity, scholars noticed that new theological approaches beyond doctrine or denominational tradition were needed (Ustorf 2008, pp. 235–36). The wall of self-reference of Eurocentric-western but also conservative-evangelical theology had to be broken (Küster 2005, pp. 184–85). European theology needed to respond to the shift from the North to the South, the expansion of charismatic churches, the emergences of migrant communities in Europe including those with a practice of reverse mission, and the unfolding of contextual theologies (Küster 2011, p. 10). New theological methods were considered, e.g., alternative forms of doing theology, such as non-Western and narrative forms (Hollenweger 1979, p. 50 in Ustorf 2008, p. 237).

The openness of Intercultural Theology meant a shift of status of theological reflection from the position of 'theological legislator' to that of 'theological interpreter' (Ustorf 2008, p. 245). Intercultural theologians from various disciplines broadened the traditional scope of their subject including religious education as academic discipline. "Intercultural Theology (ICT) is the result of a diversification of the subjects doing and practicing theological reflection. ICT reflects the newly emerging pluricentric character of World Christianity. ICT is a dynamic movement leading to the de-parochialization of Christianity, its liberation from any narrow-minded cultural captivity, which is not to lose the cultural conditions of Christian faith as such. But ICT prevents Christianity from becoming an ethnocentric tribal religion or a national religion. ICT thus is an antidote and critical conscientization process over against any cultural captivity of Christianity theology as well as any cultural superiority feeling of any kind of Christianity, i.e., it always has a strong de-colonializing hermeneutical impetus" (Werner 2021, p. 84).

### 4.3. Inner Christian Diversity as a Topic of ICT

ICT raises issues concerning the development and the current nature of Christian religious cultures that emerged from European Mission, not only in the light of postcolonial critique but also in view of the respective social, political, cultural, and religious contexts (Hock 2011, pp. 34–54). ICT also addresses the origins of African Independent Churches (AIC) that did not emerge from European Mission but have existed since Early Christianity and the quest of AIC for ecumenical fellowship (Wrogemann 2012, p. 350).

Intercultural theologians point out that ecumenism should not be restricted to the relationship between Protestantism and Catholicism and that the scope should be enlarged globally towards the plurality of confessions, denominations, and special groups. Hereby, the cultural dimension will also come to the fore (Küster 2001, p. 197). Due to the interaction between culture and religion, interdenominational interaction processes can be understood as intercultural phenomena (cf. Küster 2011, pp. 115, 132). Hence, in an interconfessional dimension of ICT relationships between Protestant Christianities are made a subject of discussion.

Fueled by an attitude of respect for the Other (Küster 2011, p. 123) and a sensitivity for plurality, ICT also defends the equal rights of the coexisting culturally diverse Christianities (Hock 2011, p. 150). Therefore, inclusion into the discourse of theology and of the Church is promoted by ICT while analyzing various forms of Christianity "from an intercultural perspective according to their particular characteristics, according to what they view as problematic, and according to their particular assumptions and priorities" (Wrogemann 2016, p. 396). By the inclusion of the position of the Other, a radical change of perspectives takes place and differences are recognized (Küster 2005, pp. 184–85). However, understanding inner-Christian differences should not come with intrusion but recognize essential non-comparability or incommensurability (cf. Sundermeier 1996, p. 13).

### 4.4. Impact of the Other

Intercultural theologians now work in various disciplines besides missiology, e.g., church history, theology of religions, ecumenical studies, and religious studies. The latter

means that not only inter-Christian, but also interreligious questions are investigated. The interdisciplinary approach also opens the definition of religion in the direction of a broader anthropological perspective. Here, religion can be seen as a cultural symbol system. Culture and religion are dialectically correlated and mutually permeated. Cultures are human-made complex webs of meaning and systems of symbols that have always been open for transcendence (Küster 2001, pp. 197–98). Consequently, what a person describes as the meaning of life and his religious identity, are culturally related manifestations. This might involve the belief in the existence of God or certain deities, specific spiritual beings, or transcendent powers that are meaningful for the person's (sub-)culture (Gebhardt 2021).

Due to globalization, ICT does not only address Churches in other continents far away. The deterritorialization of contexts and cultures is easily visible in the situation of the respective diaspora, e.g., migrant churches (Küster 2005, p. 190). The purity of the culture of origin has become highly questionable. Contexts are no longer exclusively local but have often become deterritorialized. Persons might have multiple loyalties to different groups instead of a singular centeredness on one community and their identities are no longer monocultural but hybrid (Küster 2011, p. 11). This variety of influences on a person often comes with conscious or unconscious contradictions. One of the different groups a person might belong to might be quite segregated within. A fourteen-year-old teenager living in a small town might as a confirmand visit the often very mono-ethnic and mono-lingual 10 o'clock Sunday morning service that "( . . . ) has often been cited as the most segregated hour of the week" (Fröchtling 2021, p. 112). However, the same youngster may be an enthusiastic member in a very multicultural soccer association that also plays on Sunday afternoon, hangs out with a secularized best friend who has devout Muslim parents afterwards, and on Monday evenings, chills in the local YMCA (in German: *CVJM*), he regularly meets a Nigerian mate whose father is an elder in a charismatic migrant church in a nearby city. These plural influences all have their impact on the identity formation of this youngster and make it very hard for this individual to differentiate between what is Self and Other (Wulf 2006, p. 21). This is only possible by intentional self-reflection (cf. Simojoki 2014, pp. 175–76).

Heterogeneity is characteristic for ICT. By its openness it generated a variety of theological studies. In comparison to "most expressions of classical liberation theologies dating back to the 70s and 80s, Intercultural Theology lacks a notion of uniformity that was often the background of e.g., Latin American base communities" (Fröchtling 2021, pp. 112–113).

The comprehension of the transcultural character of collective and individual forms of Christianity is a central feature of ICT (Hock 2011, p. 149). In transcultural processes cultural elements are 'translated', interpreted, re-adopted, transformed in the process, reconstituted, and developed further—and not just in an unilinear fashion, but in a multidirectional manner (Hock 2002, 70). "Today, margins and centers are in flux and so are identities" (Fröchtling 2021, pp. 112–13). Nowadays, a reflection of the personal religious identity often displays a mixture of cultural elements that have been converted in a very personal way. Within plurality, hybridity, and opposed synchronism of global and local forms of Christianity, none can claim the sole discretion of validity and interpretation. Hence, there is a need for critical reflection of this transculturality of Christianity in the tradition of enlightenment but without a rationalistic narrowing or postmodern "anything goes" (Hock 2011, p. 150). Instead, the orientation lines of Intercultural Theology are the promise of the Kingdom of God, Justice, and Peace (Hock 2011, p. 121ff). "ICT realized and spells out the four essential marks of what it means to be the church: apostolicity, catholicity, unity, and holiness—all of them together, not just one of them" (Werner 2021, p. 84).

## 5. Intrareligious Plurality within Confessional RE

### 5.1. Christian Plurality within Confessional RE

In Germany, local denominations, like Catholicism and Lutheranism, have changed considerably due to, e.g., Italian immigrants following the wave of guest workers in the 1960s, ethnic Germans from the former Soviet Union since the 1980s, and Polish EU-citizens

since 2004. The imprints of the cultural context they left behind became visible in Germany (Bauer 2019, p. 160). This contributed to the already existing intra-confessional plurality like certain social milieus, conservative orientations, or progressive styles and added new perspectives, but also caused a rise in conflict potential (Härle 2019, p. 47).

In Catholic RE and Protestant RE, differences in interpretation of presumed denominational traditions, like (not) dressing up for Sunday Service, (not) having a home altar, (not) staying for a considerable time at Church after Sunday Service, alleged confessional beliefs, the (non) observance of sanctification, (not) practicing of reverence, etc., must be constructively managed by intentionally changing perspectives. Hence, intercultural competences are also needed within denominational RE (Pemsel-Maier 2019, p. 83).

This is often overlooked since confessional RE suggests 'similarity' instead of 'diversity'. However, Christian plurality within confessional RE is on the rise. Not including the wave of persons fleeing from the Ukraine, in the first part of the 21st century approximately a quarter of all refugees were Christians. Many belong to Christian confessions other than Catholicism or Lutheranism but are, e.g., Baptists, Anglicans, or Orthodox. In German state schools, there is neither Anglican nor Baptist RE and often also no Orthodox RE. Christians with migrant backgrounds normally are enrolled in either Catholic or Protestant RE. Hence, it is highly probable these Christian youngsters who belong to Christian confessions encounter Catholic or Protestant RE teachers and pupils that are not familiar with Anglican, Mennonites, Baptist, or Orthodox traditions (Pemsel-Maier 2019, p. 88). This might be the reason for RE teachers to not pick up and further deepen utterances from, e.g., pupils of Mennonite and Baptist background (cf. Hüpping 2017, pp. 210–11).

Hence, refugees again experience a double minority position. While most pupils with a refugee background are Muslims, they are not all. However, they also differ from local Christian pupils (Pemsel-Maier 2019, p. 89). For many of them, it is new to be a believer in a secularizing society where Christianity is declining instead of growing (Pemsel-Maier 2019, p. 90; Henningsen 2022, p. 29).

*5.2. German Practical Theologians in the Field of RE Refer to the Global South*

Most German scholars in the field of religious education are not taking this interChristian kind of diversity into account. Simojoki (2017, pp. 220–31) does write about African perspectives on Jesus Christ, but while showing the potentials of these perceptions of the Global South for RE, in the subtitle of his article he refers to believers in Africa and does not mention Christians of African descent in Germany. The subtitle of the third paragraph of his article is called "Very remote and still near to us—didactical perspectives of African perspectives on Jesus Christ for RE" (Simojoki 2017, p. 228/translation ZonneGätjens). In a general German class, African perspectives on Jesus Christ would not be available in the pool of pupils' interpretations. According to Simojoki, these views come from the Global South and would be remote and alien (*fern und fremd*) for the pupils in German RE. Hence, there would be a didactical need to bring in these perspectives from outside (*von außen*) (Simojoki 2017, p. 222).

However, his work can be seen as a contribution to critically detect ethnocentrism in RE curricula, a call for renewal of RE material that still tends to concentrate on Western European contexts of Christianity and inclines to reduce plural patterns of interpretation. Simojoki might inspire those teaching Christology in considering African theologians, e.g., the Tanzanian Catholic Nyamiti (1989) and typologies of African Christologies (Stinton 2006).

Despite locating diversity far away, Henningsen (2022, pp. 260–61) showed that in often-used schoolbooks for Protestant RE, there is some attentiveness towards liberation theology in South America and its resistance against oppression and fostering of emancipation which can help to deconstruct the formula of the Global North as the sole saving instance (Henningsen 2022, pp. 260–61). The research of Henningsen also showed that—although in German RE curricula historical mission is hardly visible anymore—RE teachers in Germany are not necessarily acquainted with post-colonial education and still incline to overlook the

burdensome legacy of colonialism. Hence, they neither train the pupils to trace existing mechanisms of cultural colonialization nor do they foster skepticism and unlearning of colonial and racist representations of the Global South. Examples like contradictions of 'the civilized and developed West and the undeveloped and rural rest', calling natives as missionary objects, naming of newborns with an English forename instead of a Christian name in African language, are neither discussed nor uncovered as colonial attribution of meaning (cf. Henningsen 2022, pp. 36–38, 56, 67–68, 300–1).

*5.3. Interdenominational Plurality within Curricula for Confessional RE*

One of the preconditions for interculturalizing RE is a certain openness in the respective curricula for confessional RE. Interculturalizing RE would mean to extend and to enrich the Eurocentric confessional RE by contextual theologies in the glocal horizon (cf. Küster 2011, p. 10). Mirroring the curriculum of prospective RE teachers at university, the RE curriculum in state schools neither emphasizes topics like contextual theology nor developments in world Christianity (cf. Küster 2011, p. 10). Hence, a plea for an intercultural theological perspective while updating these curricula is necessary.

First and foremost, of course every single topic within curricula can be set and discussed from an intercultural perspective. This is not the style the syllabi are written in but there is no hindrance to implement an intercultural view on the learning field either.

Accordingly, for interculturally sensitive RE teachers, there are already possibilities within the current syllabus of the respective federal state. As the curricula only offer outlines, the RE teacher has a certain freedom. Topics and objectives in the curriculum are often fixed, but the respective learning contents mentioned in the syllabus regularly remain recommendations and are not obligations for RE teachers. Hence, an intercultural reflection of narrations in the Old and New Testament, of systematic topics like Christology, doctrine of God, pneumatology, ecclesiology, anthropology, and eschatology, and of the history of Christianity and of denominational studies is possible.

For instance, the curriculum for protestant RE at grammar school in Lower Saxony contains the topic "Church and Churches" (Niedersächsisches Kultusministerium 2017, p. 27). According to this syllabus, pupils should be able to unfold Protestant ecclesiology. Teachers are given the following suggestion how their pupils can achieve this competence, e.g., via the content for RE "ambivalence of institutionalizing" of mainline churches like the Evangelical Lutheran Church in Germany and the Roman Catholic Church versus free churches. Another option for RE teachers is to compare the Protestant and the Catholic understanding of Church and Office or to introduce them to ecumenism (Niedersächsisches Kultusministerium 2017, p. 28). Here, the door is open for RE teachers to include perspectives on ecclesiology and ecumenism from Pentecostal and charismatic migrant churches as well as world Christianities.

According to this syllabus, pupils should also be able to create perspectives for a future-proof Church (Niedersächsisches Kultusministerium 2017, p. 27). One of the suggestions for RE teachers, how their pupils can achieve this competence, is to reflect the Church in a global context (Niedersächsisches Kultusministerium 2017, p. 28). This opens the possibility to broaden the scope from global to include the glocal.

The curriculum for Protestant RE in primary schools in Lower Saxony contains the learning field "Asking questions about Faith and Church". Grade 4 pupils should be able to illustrate that the common Christian faith is lived within different confessions (Niedersächsisches Kultusministerium 2019, p. 24). RE teachers at primary schools in Lower Saxony are suggested the following content: Faith and Church in a "Protestant–Catholic–Ecumenical" perspective (Niedersächsisches Kultusministerium 2019, p. 25). While comparing both syllabi, it is noticeable that, for RE teachers in primary schools, the Protestant curriculum for Lower Saxony does not specifically restrict this learning field to the mainline churches, but rather gives a direction by naming the Protestant and Catholic confession without mentioning free churches.

In Schleswig-Holstein, the curriculum for Protestant RE at primary schools contains the competence area called "The question concerning religions in society". At the end of primary school, the grade 4 pupils, should have achieved the competence to detect similarities and differences in the religious and worldview diversity of their nearby context. The syllabus mentions "Local Jewish, Christian, and Muslim life" under "possible concretions" (Ministerium für Bildung, Wissenschaft und Kultur des Landes Schleswig-Holstein 2020, p. 24). If local Christian life also includes free and migrant churches, RE teachers might also facilitate their exploration.

From the first grade on, the Curriculum für Protestant RE in Berlin stresses that pupils should learn to observe religiously important phenomena and their expressions in their living environment (Kraft 2019, p. 21). Also, religious dialogue is mentioned explicitly with a focus on world religions and worldviews (Kraft 2019, p. 22). However, the objective for grades 3–5 to learn to express their own contexts of justification of religious topics while referring to other opinions, there is much freedom to also refer to intradenominational diversity. Even more open is the competence to able to participate substantiated in religious performance. The objective for grades 3–5 is to learn to express one's own faith while referring to other positions. The objective for grades 9–10 is to learn how to participate or to refrain from participation of religious events in a substantiated way. Protestant RE in these higher grades also aims to teach Berlin pupils to participate in different religious events, to behave in a manner that is appropriate to the situation and to be able to relate to their own form of expression (Kraft 2019, p. 23).

### 5.4. Intracultural Plurality According to Staff of Confessional RE

As an outlook, RE teachers of Protestant RE with a multicultural biography were asked how they are currently interculturalizing RE and what kind of strategies they have for further intercultural opening of denominational RE in German state schools.

Ghanaian Peter Arthur came to Germany as a young adult thirty years ago. He is a pastor of an intercultural Free Church in Berlin, but also teaches protestant RE in a primary state school in different age groups.[19] During his BA studies, among others, Mr Arthur took modules in "Intercultural dimensions of the Bible", "History of Christion Mission", "Christian confessions", "World Religions", and "African Traditional Religions". During his MA studies, Mr Arthur absolved modules like "Processes of Translation, Inculturation and Intercultural Communication", "Christianity in an Intercultural Perspective", "Religions, Churches in Europe and the Western World", and "Religions, Churches and Theology in Africa". In his parttime work at the primary school, Mr Arthur teaches what he calls "classic Lutheran" RE.[20] His colleagues in the RE department of this primary school all have a German background. In his RE, about one third of the children have a migration background and there are "only very few pupils" who are members of the Lutheran Church in Germany. Most Christian pupils are Catholic, and a few are Methodist. In his primary school, Mr Arthur appreciates "the openness to teach children of different denominational backgrounds together. So, I also view their concept to be kind of open, already. Thirty years ago, this would not have been possible in Germany."

Mr Arthur has developed practices to actively identify and deepen contributions of pupils of different Christian backgrounds. He tries to explain in class "the varieties of theologies mirrored in my pupils' answers. I don't really have developed a method to do so, but rather intuitively ask them questions when I realize a certain denominational thinking included in their contributions. Then I will point this out and explain it to the class." This requires pre-knowledge Christian confessions. In Mr Arthur's case, he acquired this information during his academic education.

Paul Adeyemi teaches protestant RE at a primary school in a small town in the North of Germany.[21] During his MA studies, Mr Adeyemi also absolved the modules "Processes of Translation, Inculturation and Intercultural Communication", "Christianity in an Intercultural Perspective", "Religions, Churches in Europe and the Western World", and "Religions, Churches and Theology in Africa". He was baptized in the Catholic Church as

a child, but "decided to become baptized again as a teenager after experiencing something like a spiritual awakening in the USA".[22] Most of his RE colleagues at school have a German and Protestant background. The person teaching Catholic RE has a German background as well. Around 5% of the participants of his RE have a migration background. Mr Adeyemi tries to "encourage exchange of opinions and thoughts. However, most children are either belonging to the protestant church or are not considering themselves part of a church. I think the biggest issue in RE today is not pluralization but rather secularization. Many children say they do not believe in God" (see Note 22). Mr Adeyemi's awareness on this matter might derive from his former training as within the Module "Religions, Churches in Europe and the Western World" the diminishing influence of religion on state and on civilians in Germany was a major topic. Mr Adeyemi has developed teaching practices that fit the secularized and—in comparison to Mr Arthur's classes—less multicultural groups. He tries to bring the diversity of cultures, confessions, and religions of his small town and the region around it to the class. "I think, interreligious learning and intercultural learning works mostly by way of encounter. In contexts where the student body is diverse, providing opportunities for exchange and intercultural encounter is helpful. Where diversity is not present, one can try to artificially create it, for instance through digital encounters or field trips" (see Note 22).

RE teachers like Mr Arthur and Mr Adeyemi can be seen as an inspiration and as a leading force for the teaching staff when it comes to include intercultural theological perspectives and glocalization within protestant RE. During their academic studies, they were well trained on intercultural and interreligious subjects. Their open-eyed vision on cultural diversity also within the protestant denomination makes them aware how to react if relevant topics are brought to the fore by pupils and can proactively plan intercultural glocal encounters digitally or in person to enrich their classroom practice as well.

Their stories show that there are Protestant RE teachers furthering an interculturalizing approach. However, currently there is no statistical evidence for the general degree to which Protestant RE is or is not meeting the aims of interculturalizing education.

Addressing local and regional diversity of Christian Churches and confessions during RE is also what the Chamber for Worldwide Ecumenism of the Lutheran Church in Germany (*Kammer der EKD für Weltweite Ökumene*) is recommending to all RE teachers. In its publication it refers to the approximately 3000 international congregations in Germany. However, the interculturalizing of RE is not a 'mission completed'. The Chamber for Worldwide Ecumenism is concerned that not all staff that are linked to the Protestant Church are as interculturally trained, equipped, and ready to meet the challenges that diversity within Protestantism brings like Mr Arthur and Mr Adeyemi are. Indeed, those two teachers might be the exception to the rule—both when it comes to the modules they absolved during their training as well as their personal cultural and religious biographies.

Indirect evidence that the two profiled teachers might be the exception rather than the rule comes from this Chamber. The Chamber states: "New challenges arise concerning the ecumenical-theological competence of persons who commit themselves or are employed in Church contexts. For those who are preparing themselves for a service in protestant Churches or Protestant RE in Germany, a basic knowledge of orthodox, Catholic, free church like Baptist and Methodist as well as Pentecostal traditions is equally vital as basics concerning global forms of Christianity, i.e., in Africa or Asia" (Kammer der EKD für Weltweite Ökumene 2021, p. 7—Translation EZG). The Chamber urges to embed contents and practical orientation concerning ecumenical plurality within the academic study and training for ministry and teaching positions in the congregation, diaconal services, and schools in intercultural and interreligious cooperation (Kammer der EKD für Weltweite Ökumene 2021, p. 8). The Chamber emphasizes that interculturality of worldwide ecumenism affects the different confessions as well as the cultural plurality within confessions. Hence, intercultural theological focus in the academic training for all staff should lie on the culturally shaped diversity of Christian witnesses (Kammer der EKD für Weltweite Ökumene 2021, p. 9).

The prescriptions for changing the scenario and making the two profiled teachers the rule and not the exception might lie in including Intercultural Theology in all university education programs for teacher trainees. The mission of interculturalizing RE will profit from RE teachers with broader knowledge on global and glocal Christianities. More diverse seminars during teacher training at respective theological faculties can be arranged by better faculty support within the faculty of Lutheran Theology and/or the possibility to include modules at fellow faculties for Islamic, Jewish, or Catholic Theology, or institutes of religious studies.

In Lower Saxony, the Lutheran and Catholic Churches have indicated that they are planning to convert their cooperative RE—that is in principle still a separate confessional RE but invites pupils by turns to take part in RE of the other denomination—into a Christian RE in joint responsibility of these Churches. Should this Christian RE become a reality, they call for changes in the university education for prospective RE teachers. The cooperation between Lutheran and Catholic faculties within the university but also with other universities should be intensified. They request university staff to address confessional heterogeneity in all theological modules and to introduce religious studies modules into teacher training (Schulreferentinnen und Schulreferenten der evangelischen Kirchen und katholischen Bistümer in Niedersachsen 2021, p. 37).

In conclusion it can be stated that the Mission of Interculturalizing RE is "to be continued ... ".

**Funding:** This research received no external funding.

**Institutional Review Board Statement:** Not applicable.

**Informed Consent Statement:** Informed consent was obtained from all subjects involved in the study.

**Conflicts of Interest:** The author declares no conflict of interest.

## Notes

[1]   In the 20th century, Church Tax is approximately 9% of the income taxation.

[2]   In the 21st century, this compensation is still paid—on top of the church taxes to both Catholic and Lutheran Churches.

[3]   Landeskirchliches Archiv Stuttgart, D1/78; https://de.evangelischer-widerstand.de/html/view.php?type=dokument&id=100 (accessed on 22 March 2022).

[4]   BVerwG, Urt. V. 32.2.2005.-6 C 2.04.

[5]   Hence, in several federal states, the Turkish governed DITIB communities in Germany are not allowed to carry out Islamic RE in German public schools (Bauer 2019, p. 99).

[6]   If there is proof of shortage of RE teachers, a person with an equivalent degree can teach parttime for a permit of one school year that can be prolonged if necessary. In this case, there is no *vocatio* but a restricted church permission to teach RE that year.

[7]   In most Lutheran Regional Churches (*Landeskirchen*) members of certain denominations of the Working Group for Christian Churches *ACK* can receive the *vocation.* In Lower Saxony, teachers who are members of the reformed, Methodist, or independent Lutheran Church or the Moravian Congregation can teach RE if they have the *vocatio* of the Lutheran Church of Hannover. Moreover, in exceptional cases, teachers from other free churches can teach with a certificate of non-objection of the Lutheran Church of Hannover after hearing of the application. Some RE teachers in Germany enter a double church membership to obtain the *vocatio.* E.g., next to the membership of a congregation that does not belong to an *ACK* Church, they decide to stay within their non-*ACK* congregation but also become a member of the Lutheran Church. As not all migrant churches are on the *ACK*-list, this is a regular option to be granted the *vocatio*.

[8]   Teachers at public schools that were given the status of civil servant—including most of the religious educators—normally have the competence and qualification to teach not only RE, but also a second or a third subject. Hence, if they decline to teach RE anymore, they might still teach the other subjects.

[9]   Gemeinsame Vokationsordnung, §3, Abs. 2.

[10]   Codex Iuris Canonici 1983 Can. 804 §2.

[11]   *Grundsatzurteil* BVerfGe 74, 244 (252).

[12]   BVerfGe 74, 244 (252).

[13]   For instance: Mennonite RE and RE of the reform movement Ahmadiyya Muslim Jamaat.

[14]   Currently, approximately 75% of Muslims living in Germany follow Sunni Islam.

[15] The pupil will receive a 'guest status' upon application. One plausible reason is that RE of his/her own religious community does not take place, because lack of the respective confessional teachers or confessional pupils. If the application is granted by the 'host' religious community, its mono confessional RE with this 'guest' does not become a cooperative RE (see Section 1).

[16] BVerfGE 74, 244.

[17] Section 7 (1) HmbSG.

[18] www.schulrechthamburg.de/jportal/portal/bs/18/page/sammlung.psml?doc.hl=1&doc.id=VVHA-VVHA000000167&documentnumber=1&numberfresults=1&showdoccase=1&doc.part=F&paramfromHL=true (accessed on 16 March 2022).

[19] Mr Arthur also teaches an interdisciplinary subject at a reform pedagogical secondary school. This also is a parttime position.

[20] Self-statement of Peter Arthur on 30 March 2022.

[21] This RE teacher chose his own pseudonym.

[22] Self-statement of Paul Adeyemi on 29 April 2022.

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
