# Peer review of "Interculturalizing Religious Education—Mission Completed?"

_religions, doi:10.3390/rel13070653_

Round 1
Reviewer 1 Report
See attached.

Author Response
Thank you for your critical but plausible review.
The status quo of the interculturalization of protestant RE was my orginal focus point. However, after sending the abstract, I was advised to consider non-German readers who are not familiar with the (historically grown) set-up of RE.
Taking that advice maybe too seriously, the historical part has grown and grown. Hence, I agree that as a result the balance is missing.
I will try to re-structure the article.
Reviewer 2 Report
The article constitutes a fascinating and detailed account of the handling of religious difference in German religious education, and engages thoughtfully with the issues involved in an interculturalisation of the curriculum subject. It will be valuable to those with an interest in comparative studies of religious education across the education systems of different nations, and to those interested in developments in the delivery of RE in societies that are both religiously plural and secular.
I would suggest some minor revisions before publication.
i) Further proof-reading/language-editing is needed. There is an occasional missing word or spelling error (eg 21rst appears more than once) and some clumsiness in vocabulary and syntax that could be corrected by a native English speaker or someone with equivalent knowledge of the English language.
ii) The article would benefit from an introductory paragraph explaining briefly where the article has come from and where it is going.
iii) The first part of the article involves an historic overview of German religious education from the Middle Ages to the present. The coverage of this education is qualitatively different between the different eras, that of the earlier periods being very superficial while there is more detail for the later periods. This unevenness wouldn't be such a problem if it was made clear that the purpose of the earlier sketchy historical runthrough was merely to provide a contextual foundation for the more detailed presentation of later times. I suggest that the earlier periods are incorporated under a 'before' or 'pre' heading (such as 'German religious education pre/before 1945?/the 1960s? or whatever starting point seems appropriate') and that this section ends by drawing out from the historical legacy the key aspects that are pertinent to, and have a continued impact on, recent, current and future developments.
iv) The discussion of qualitative empirical research data from Mr Arthur and Mr Adeyemi sits uncomfortably with the rest of the article and its broader focus, and the evaluative, normative paragraph ll 817-823 is out of keeping with the style. It is the sort of material that might be expected in a different kind of article. For this reason I suggest removing the section on this research altogether or, if it is considered to have a useful contribution to make to the broader picture, just including a brief reference to any already existing publication of these research findings on the level of a secondary source for the present article.
Author Response
Thank you for your review, your suggestions for headings and language-editing.
Your critic is plausible. The passage about Mr Arthur and Mr Adeyemi is indeed written in a different style. Unfortunately, there is no earlier publication I can refer to. However, I will try to highlight their stories in order to make evident that their strategies in everyday teaching show that they are fostering the interculturalization within protestant RE.
Round 2
Reviewer 1 Report
The addition of an introduction structures the article well. The reader now knows the argument that is to be developed. Re-structuring the article and pulling section 3 out of the general history section is also helpful, because it points to interculturalism, or shows existing attempts to go beyond simple mono-confessional RE. Review the section numbers: line 534 should be 3.4, not 2.4. On line 599, the section should be 4.2, and on line 651, the section should be 4.3.
The conclusion is a bit thin. For example, there is no summary of the argument and conclusion. But a more robust conclusion is not a deal breaker in this instance.
Author Response
Thank you very much for your review. It was very helpful.